# CRISPRbuilder-TB: *"CRISPR-builder for tuberculosis"*. Exhaustive reconstruction of the CRISPR locus in *mycobacterium tuberculosis* complex using SRA

**Christophe Guyeux**[1]*, **Christophe Sola**[2,3], **Camille Noûs**[2], **Guislaine Refrégier**[4]

**1** FEMTO-ST Institute, UMR 6174 CNRS, DISC Computer Department, Univ. Bourgogne Franche-Comté (UBFC), Besançon, France, **2** IAME, UMR1137 INSERM, Université Paris, Université Paris Nord, **3** Université Paris-Saclay, Saint-Aubin, France, **4** Ecologie Systematique Evolution, Batiment 360, Université Paris-Saclay, CNRS, AgroParisTech,Orsay 91400, France

* christophe.guyeux@univ-fcomte.fr

**Data Availability Statement:** All relevant data are within the manuscript, its Supporting Information

## Abstract

*Mycobacterium tuberculosis* complex (MTC) CRISPR locus diversity has long been studied solely investigating the presence/absence of a known set of spacers. Unveiling the genetic mechanisms of its evolution requires a more exhaustive reconstruction in a large amount of representative strains.

In this article, we point out and resolve, with a new pipeline, the problem of CRISPR reconstruction based directly on short read sequences in *M. tuberculosis*. We first show that the process we set up, that we coin as "CRISPRbuilder-TB" (https://github.com/cguyeux/CRISPRbuilder-TB), allows an efficient reconstruction of simulated or real CRISPRs, even when including complex evolutionary steps like the insertions of mobile elements. Compared to more generalist tools, the whole process is much more precise and robust, and requires only minimal manual investigation. Second, we show that more than 1/3 of the currently complete genomes available for this complex in the public databases contain largely erroneous CRISPR loci. Third, we highlight how both the classical experimental *in vitro* approach and the basic *in silico* spoligotyping provided by existing analytic tools miss a whole diversity of this locus in MTC, by not capturing duplications, spacer and direct repeats variants, and IS*6110* insertion locations. This description is extended in a second article that describes MTC-CRISPR diversity and suggests general rules for its evolution. This work opens perspectives for an in-depth exploration of *M. tuberculosis* CRISPR loci diversity and of mechanisms involved in its evolution and its functionality, as well as its adaptation to other CRISPR locus-harboring bacterial species.

## Author summary

In this article, we tackle the bioinformatical issue of the reconstruction of the *Mycobacterium tuberculosis* complex CRISPR locus using short read sequences without requiring genome assembly. We first show that many complete genomes, as found in public

files, and the Git repository https://github.com/
cguyeux/CRISPRbuilder-TB.

**Funding:** The author(s) received no specific
funding for this work.

**Competing interests:** The authors have declared
that no competing interests exist.

databases and often reconstructed by *de novo* assemblies, often contain errors on this
locus as well as on other repeated sequences. We provide an in-depth description of our
new method, designated as 'CRISPRbuilder-TB', and we show that our method provides
much more exhaustive and reliable information (on DR variants, spacer diversity, global
structure) than Crass and CRISPR_detector. The new and unsuspected genomic diversity
we detected is described in a companion paper. Scripts are available to adapt the tool to
other species.

This is a *PLOS Computational Biology* Software paper.

## Introduction

The CRISPR locus of *Mycobacterium tuberculosis* complex (MTC) the agent of tuberculosis
(TB) was first described in 1993 under the "Direct Repeat" locus designation [1,2]. It is made
of 36 nucleotides-repeats interspaced by unique spacers of a mean of 37nt (interval: 25-45nt).
The repeats were soon themselves designated as *Direct Repeats* and abbreviated as such (DR),
and the sequences of one unique spacer + one DR were called Direct-Variant Repeats (DVRs).
The two first sequenced isolates (*M. tuberculosis* H37Rv and *M. bovis* BCG) gave access to 43
different spacer sequences. The detection of their presence/absence led to the development of
the innovative "spoligotyping" method [3]. This method became very popular by its ease of
implementation and digital format, and was indeed instrumental to decipher the global popu-
lation structure of MTC [4]. More recently, Whole Genome Sequencing (WGS) studies indeed
confirmed that for the 6 main human lineages (L1 to L6) and many sublineages, the spoligo-
typing signature allows an approximate taxonomical assignment [5]. Still some generic signa-
tures remain either meaningless, imprecise or convergent, thus largely justifying the use of
SNPs as preferred taxonomical markers either globally [6], or for L4 [7], L1 [8], or L2 [9].

As in other species with functional CRISPRs, this locus is accompanied by a set of CRISPR
associated (*cas*) genes. Their number and nature make MTC CRISPR type fall into Type III-A
group inside CRISPR-Cas taxonomy [10]. CRISPR-Cas locus was recently shown to be active
in H37Rv [11]. Yet, part or the entire region is deleted in several MTC sublineages [12].
Whether the deletion of some of the *cas* genes in the CRISPR-Cas locus may promote genomic
instability in some epidemic strains of MTC is another important question [13].

The genomic diversity of the CRISPR locus has been investigated in detail as early as 2000
in a study by J. van Emdben *et al.* showing that spacer duplication, spacer variation, and
IS*6110* insertion sites could be found in the various phylogenetic lineages of MTC [14]. How-
ever, it concerned a very small sample (n = 34) and did not include any investigation on the
*cas* genes [15,16]. Understanding evolutionary dynamics of this locus now requires the explo-
ration of CRISPR-Cas region on an extended dataset.

The classic *in vitro* approach to spoligotyping lists the presence or absence of a well-known
list of spacers in a sample. This robust method has been largely applied *in vitro* [3]. This
approach however did not allow the exploration of many characteristics such as if the order of
the spacers is different in one strain or the other. Neither did it reveal if there had been a dupli-
cation of part of the locus. Finally, it did not provide information on the presence of insertions
such as IS*6110*, nor on the existence of single nucleotide polymorphism (SNP) in its direct
repeats or spacers. This masks potential functionally significant changes in the loci, and makes

it impossible to carry out thorough and more *in-depth* evolutionary studies. New *in silico-based* approaches (SpolPred, SpoTyping) were developed to produce spoligotypes based on genome reads [15,17]. Even if these methods similarly reveal the presence/absence of the spacers, they have the same limitations as *in vitro* spoligotyping techniques.

A more exhaustive way to explore WGS data is by traditional *de novo* assembly tools such as Velvet [18]. The reconstruction is however much more difficult for the CRISPR locus. Indeed, in this part of the genome, the same DR sequence is found between each pair of spacers. Since the size of a DR is not far from that of the k-mers usually used during assembly, there is a risk of wrong bifurcations when searching for an Eulerian path in the De Bruijn graph associated with this assembly.

Methods were proposed to describe and/or reconstruct CRISPRs: Crispr Recognition Tool (CRT) upgraded in CRTmeta, CrisprFinder that is a CRT-based tool with user-friendly interface and that was upgraded in CrisprCasFinder, PILER-CR, Crass, CRISPRdetector, meta-CRISPR) [16,19–23]. However, each of these methods suffers from specific or common limitations we needed to circumvent to exhaustively address the genomic dynamics of TB CRISPR. Indeed, CRT and CRISPRCasFinder operate until now solely on assembled genomes, Crispr Recognition Tool for metagenomes (CRTmeta) is very slow with reads archives, Crass requires relatively long reads (more than 100bp) and provides maps that may be difficult to interpret with frequent missing or erroneous edges [21], CRISPRdetector only indicates whether the sequences contain a CRISPR and the corresponding DR sequence, metaCRISPR only uses paired-end reads and can only resolve insertions in the size of the insert chosen during the sequencing procedure (see **S1 Table** for links to each tool and a short description of their objectives and outputs). To the best of our knowledge, these tools are unable to detecting and/or resolving duplications and/or presence of Insertion Sequences.

In this article, we present a new general procedure termed 'CRISPRbuilder-TB' to reconstruct *Mycobacterium tuberculosis* CRISPR-Cas loci from short reads under a semi-automatized process. CRISPRbuilder-TB is inspired by previous developments [24,25] and particularizes the De Bruijn approach to the specific case of reconstructing CRISPR loci based on SRA, with the condition that its standard DR sequence is known. Applied on TB SRA without any genome assembly step, CRISPRbuilder-TB proved reliable and robust with reads of more than 75bp. The remarkable elements that were found in this locus by MTC lineages is the subject of a separate publication [1]. We emphasize its usefulness by showing that close to 50% of MTC complete genomes available in public NCBI database exhibit unreliable CRISPRs, and by detailing the different kind of evolutionary events that can affect CRISPRs, at least in MTC genomes.

## Results

### Description of CRISPRbuilder-TB

A known limitation to CRISPRs reconstructions is the difficulty to handle large sets of reads. As our aim is to reconstruct a CRISPR whose common DR is known, the first step of CRISPR-builder-TB is to filter the SRA. We kept not only reads similar to DR sequences but also reads similar to other sequences that may be linked to the CRISPR (**Fig 1** step 1). We call this catalogue of sequences linked to CRISPR, the "catalogue of remarkable sequences". This catalogue is thereafter also aimed to express the composition of a definite CRISPR in a self-explanatory way, referring to known DR and spacers. We thus aimed at making this catalogue as exhaustive as possible. To do so, we explored a large and representative set of SRA from MTC strains (Dataset #3, **Fig 2**, n = 434, L1 to L6, animal strains and *M. canettii*). This method did not lead to the identification of new spacers in MTC *stricto sensu* (excluding *M. canettii*). It identified

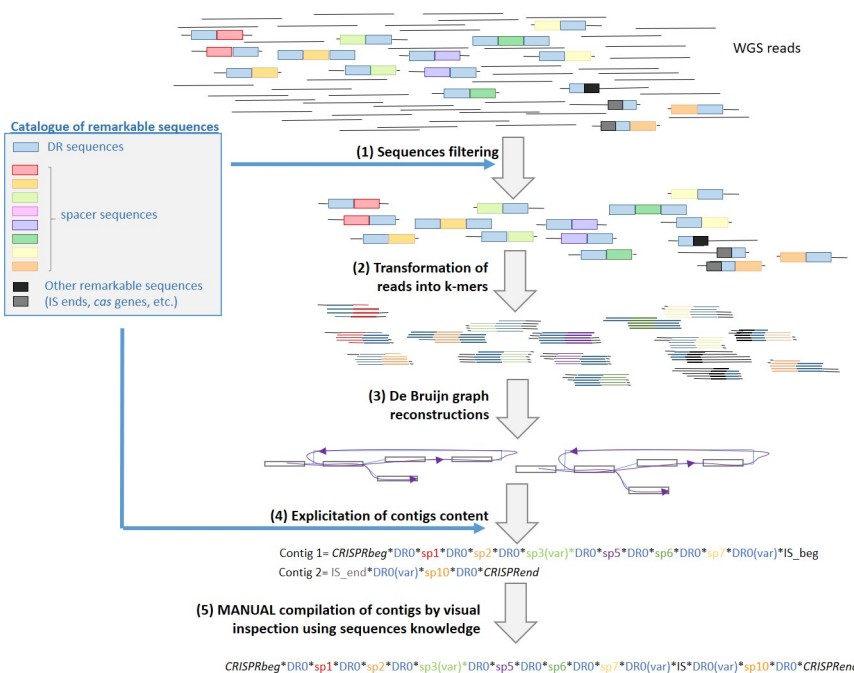

**Fig 1. Diagram showing the process of MTC CRISPR Locus reconstruction.** All steps were performed using tailored python scripts embedded in CRISPRbuilder-TB. Step 1: WGS reads filtering by blasting the reads with the catalogue of remarkable sequences (threshold e-value = 1e-7). Step 2: transformation of reads into k-mers which length was adjusted to read length n (k-mer = 4/5n). Step 3: De Bruijn graph reconstruction implemented in python (see **S2 Text**, part b). Step 4: Replacement of known sequences by their annotation as per the catalogue of sequences of interest. Rare unknown sequences are kept as such for manual inspection. Step 5: manual reconstruction of full CRISPRs using the available contigs (mostly concatenation according to IS*6110* recovered ends).

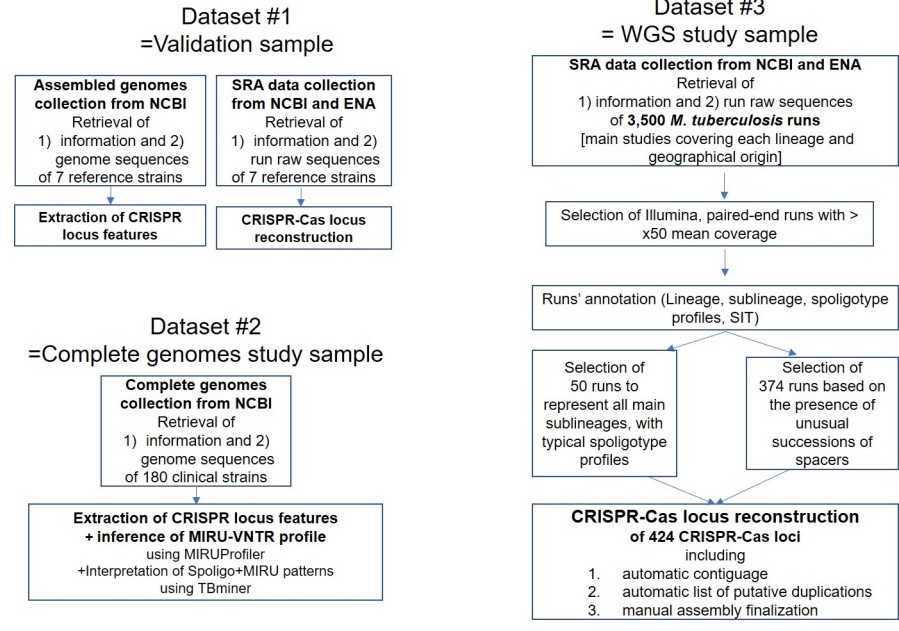

**Fig 2. Datasets used in this study.**

rare variants among the 68 spacers from MTC *stricto sensu* (n = 20; max = 2 SNPs per spacer, **S2 Table**) and several DR variants (n = 28; max SNPs per DR = 3, **S2 Table**). These procedures recovered other remarkable sequences: IS*6110* borders, and borders of the locus, that appended the catalogue of remarkable sequences. We finally added partial sequences of all *cas* genes.

CRISPRbuilder-TB *per se* is a tool that builds contigs from reads using k-mer extension in a process that is similar to CRT [16] (**Fig 1** step 2) and orders them using a De Bruijn graph reconstruction (**Fig 1** step 3). The resulting number of contigs depends on the number of non-tandem duplications and the number of IS*6110* sequences in the locus, each of this event increasing the number of contigs by one. The specificity of CRISPRbuilder-TB is then to replace known subsequences into strings of words using the MTC CRISPR remarkable sequences catalog described above (**Fig 1** step 4). This replacement allows fast manual inspection of the contigs, highlithing potentially compatible contig ends and unknown sequences (new variants and/or IS*6110* insertions, **Fig 1** step 5). It allows to rebuild the global structure of the locus even if it includes large insertions, deletions or duplications.

## Reconstruction of simulated CRISPR

We simulated a first set of three MTC CRISPR profiles with random IS*6110* insertions, deletions, duplications, and rare single nucleotide mutations and the corresponding reads. The simulated CRISPRs carried spacers variants, IS*6110* insertions, DVR duplications (**S1 Text** for exhaustive description of the three simulated CRISPRs and reconstruction by the different tools, **S1 Fig** for simple visual display of the same reconstructions, and **Table 1** for a summary of a set of interest).

We then tried to reconstruct the locus based on simulated reads. This was done with a first set of CRISPR simulated with relative high rate of duplications (**Table 1**) and a second set with a lower duplication rate (**S1 Text**, **S1 Fig**). In the two sets, CRISPRbuilder-TB outperformed Crass in reconstituting CRISPR as it could do it with short reads, with exact identification of flanking regions, explicit replacement of sequences by annotation which enabled easy full reconstruction based on the first set of contigs, and with no ambiguity (absence of multifurcations) (**Table 1**, **S1 Text**, **S1 Fig**). In contrast, Crass efficiency was impacted by DVR duplications (failure in reporting a spacer that is duplicated, sp26 in CRISPR-2 (**Table 1**), and mistook spacers for flanking sequences while dealing with longer reads (CRISPR-3 or CRISPR-6, **S1 Text**, **S1 Fig**). Of note, CRISPR_detector failed in identifying the CRISPR for reads of 125 and 300 bp. When able to process the reads, as presented in **S1 Table**, it gives out only very limited information: only consensus DR sequence. Altogether, CRISPRbuilder-TB achieved perfect reconstruction of all loci: independently of the read length and of depth, CRISPRbuilder-TB retrieved all the DVR from the simulated profiles in the correct order, and included no false DVR (precision = 1 and recall = 1, n = 80).

We also tried to implement metaCRISPR tool. However, this tool did not provide any DR and spacers when applied on CRISPR-1 and could not perform the different steps of the pipeline for the two other ones.

## Evaluation of CRISPR locus reconstruction based on WGS data of MTC reference strains with complete genome sequences

We then used CRISPRbuilder-TB to reconstruct the CRISPR loci of the seven best MTC studied strains for which complete genome sequence is available, using corresponding sequencing runs. This selection of strains covers a large MTC diversity (four L4 strains, two *M. bovis* BCG variants, and one L2 strains; **S3 Table**).

**Table 1. Tool comparison for the reconstruction of three simulated MTC-like CRISPRs.**

| | Simulated evolved TB-CRISPR-1 | Simulated evolved TB-CRISPR-2 | Simulated evolved TB-CRISPR-3 |
|---|---|---|---|
| *Characteristics of the simulated CRISPR sequence* | | | |
| Total spacer number | 70 | 68 | 70 |
| Total IS number | 2 | 3 | 1 |
| Spacer variants | 4 (sp11, sp33, sp51, sp65) | 2 (sp7, sp52) | 4 (sp13, sp26, sp30, sp60) |
| DR variants | 7 (DR2, DRb2, rDRa1, DRb1, DR6, DR4, DR5) | 7 (DRvar*, DRb2, rDRa1, DRb1, DR25, DR4, DR5) | 7 (DRvar*, DR2, rDRa1, DRb1, DRvar*, DR4, DR5) |
| Duplications | DVR35, DVR48, DVR67 | DVR35 | DVR35, DVR37 |
| IS Insertions | standard at sp34, plus one after sp61 | standard at sp34, plus one after sp25, one after sp55 | standard at sp34 |
| Deletions | DVR62 | DVR26 | none |
| *Characteristics of the reads derived from the simulated sequence* | | | |
| Read length used | 75 bp | 125 bp | 300 bp |
| *Characteristics of the CRISPR loci reconstructed by different methods* | | | |
| **Output Crass** (Skennerton et al, 2012) | no CRISPR | 1 CRISPR with DR0, 3 contigs, 3 multifurcations in the reconstruction diagram, 65 spacers (lacking sp26, 55 and 56), flanking regions = sp55 and IS | 1 CRISPR with DR0, 2 contigs, 5 multifurcations in the reconstruction diagram, 60 spacers lacking sp2, 21, 24, 25, 30, 46, 47, and 58), flanking regions = sp2, sp46, sp47, sp58, sp21, sp25, sp24, sp30. |
| **Output CRISPR_detector** (Ben Bassat et al, 2015) | Confirmation of 1 CRISPR. Correct sequence of DR0 but none of the DR variants identified. No information on spacers. | no CRISPR found | no CRISPR found |
| **Output CRISPRbuilder-TB** (this study) | | | |
| Automatized output | 70 spacers, 23 contigs, 4 sections of unknown sequences (later identified as spacer variants) | 68 spacers, 4 contigs, 3 sections of unknown sequences (later identified as 2 spacer variants, 1 DR variant) | 70 spacers, 2 contigs, 5 sections of unknown sequences (later identified as 3 spacer variants, 1 spacer+DR variant, 1 DR variant) |
| Output after manual inspection | **100% correctly decoded sequence**: 1 contig, identification of spacer variants, deletion, duplications, IS insertion | **100% correctly decoded sequence**: 1 contig, identification of spacer and DR variants, deletion, IS insertions | **100% correctly decoded sequence**: 1 contig, identification of spacer and DR variants, duplications |

CRISPR reconstructed by CRISPRbuilder-TB were identical to those of the complete genome sequences (**S3 Table, lower part**). In addition, it provided more extensive information than that provided by the Spolpred (**S3 Table, upper part**). Indeed, we detected the DR variants found between spacers 25 and 26 (truncated version), between spacers 30 and 31, between spacers 64 and 65, 66 and 67, and between spacers 67 and 68 as described previously [14]. We also identified the expected IS*6110* sequence in the DR between spacers 34 and 35. Last, we detected a duplication of spacer 35 and the adjacent DR (Direct Variant Repeat 35 or DVR35) as described by van Embden *et al.*, but we always identified it at the 3' end of DVR41, not DVR45 as described in text by these authors [14]. At the level of the spacer variants, a single discrepancy was identified around spacer 13 in H37Ra: in the complete genome, there is a variant of the spacer with 10 more nucleotides, corresponding to an unexpected tandem duplications of nucleotides. This spacer is surrounded by two distinct variants of DR, one with a size 46 and the other with a size 39 and these unlikely DR inflations again correspond to tandem nucleotide duplications. The very unusual patterns affecting these sequences make them doubtful.

Altogether, the CRISPR-Cas loci reconstructed by our pipeline using WGS of these reference strains match perfectly with the public complete genome sequences. This validates our analytic pipeline to annotate and reconstruct CRISPR-Cas locus based on short-reads runs.

## CRISPR region in other MTC complete genomes

To explore the reliability of CRISPR regions of complete genomes available in Public databases, we extracted the CRISPR-locus from all the additional MTC complete genomes we identified at the time of the study (n = 187, sample #2, **Fig 2**). According to the SNPs that they carried, these genomes belonged to L1 (n = 6), L2 (n = 44), L3 (n = 2), L4 (n = 112), L6 (n = 2), L_bovis (n = 15), unknown(mic+can) (n = 6). None belonged to L5. At the IS*6110* level, all complete genomes had the common insertion downstream of spacer 34. Only another IS*6110* copy was identified, in front of spacer 46 of strains of sublineage L2.2.1.

Most sequences seemed of high quality according to the presence of known CRISPR remarkable sequences and to the presence of compatible Coll *et al.* SNPs for classification [6], such as L4, L4.1, L4.1.2, L4.1.2.1 for tub_LN3589 (**S4 Table**). However, 25 genomes (~14%, for instance EAI5_NITR206, CAS_NITR204, GG-77-11) accumulated multiple variations of spacers and DRs, at sizes varying greatly, making their quality dubious. For example, strain GG-77-11, lineage 4.3.2, had mutations in spacers 19, 20, 21, 25, 32, 34 and 42.

We also noticed a low frequency of strange patterns. These patterns do not match previous knowledge: the 27 complete genomes belonging to L4.1 or L4.2 did not include spacer 35 after spacer 41; an additional set of eighty-six genomes exhibited repetitive sequences patterns that were likely contradictory with their classification according to SNPs (**S4 Table**). This was particularly made clear when identifying the main lineage corresponding to their repetitive sequences using TBminer [26] (**S5 Table**). In many instances, the H37Rv reference genome sequence seems to have been used as a default sequence, jeopardizing the quality of the assembly. This impedes their use to explore CRISPR diversity and infer evolutionary dynamics of this highly important locus.

## CRISPR evolutionary events in MTC

We reconstructed the CRISPR-Cas locus of 434 strains representing the diversity of the MTC lineages and showing interesting features (**Fig 2**, sample #3). As explained previously, these results and lessons is the subject of the companion article [1]. Yet we want here to highlight some of the patterns we observed that may be observed in the CRISPRs of other species.

First, we can say that the global CRISPR profiles obtained are validated by their consistency with SNPs-derived lineages and sublineages. For example, the deletion of spacers 43 to 50 in the extensive nomenclature (a part that corresponds to spacers 33–36 in the 43-spacers spoligotype patterns) was found in L4 samples, etc., (**S6 Table**, **S7 Table**).

Second, we examined the main evolutionary events that occurred in our subsample.

Regarding DR diversity, as expected the reference version of the DR, called DR0, is largely predominant. And if we except the DR located between spacers 25 and 26, almost all DR variants have the same size, *i. e.* 36 nt. Some DR variants observed in the 7 references strains were confirmed in all samples. Namely, the same DR variants were always found respectively between spacers 30 and 31 (DR2), between spacers 66 and 67 (DR4), between spacers 67 and 68 (DR5), and between sp64 and 65 (DR6, **S8 Table**). In addition to these pervasive variants, we identified that some variants were specific to lineages or sublineages: all L1 samples, and only they, have a DR3 variant between spacers 50 and 51, and those of sublineage L1.1.1.1 have DR1 between spacers 14 and 15 (**S8 Table**).

Other punctual mutations occurred in spacers: the strains of Africanum L6 lineage carry a variant of spacer 4, the L7 ones (Ethiopian-Abyssinia Lineage) all have a variant of spacer 6. The ERR234156 corresponding to a strain from lineage 1.1.1, as well as other strains from this sublineage, has a spacer 38 variant (**S7 Table**). Of note, the spacer variants had up to 2 SNPs compared to reference sequence.

We also observed duplications in the reconstructing CRISPR: 1) a large duplication including DVR14 to 20 between spacers 20 and 21 in sublineages 1.1.1.7 and 1.1.1.8; 2) a tandem duplication of spacer 29 is found in sublineage 1.1.3, as well as spacers 5 and 21 in L3; 3) a large duplication of 25 DVR (starting with DVR7) between spacers 57 and 58. We also confirmed that there is duplication of spacer 35 in every strains between spacers 41 and 42, with the notable exception of sublineages 4.3 to 4.9 (**S7 Table**).

Finally, we observed IS*6110* additional to the well-known copy between spacers 34 and 35. They were found in the sense or antisense direction and were often at the border of deletions (**S7 Table**).

## Computational speed

While performing these tests we explored the speed of the automatized part of our tool on a standard machine with six cores of 2.8 GHz and 32 Go of RAM. We measured computational time for 15 SRR representative of MTC diversity, of read lengths (from 75 bp to 249 bp) and average coverage (70,3 to 723,7). The maximal time was 1038 seconds per SRR (17 min). Time correlated poorly with both read lengths or coverage ($R^2 < 0,05$), but correlated reasonably with the number of reads ($R^2 = 0,2665$, p = 0.024, **Fig 3**). Another important determinant of computational time was the number of spacers in the final locus. One proxy for this number is the lineage to which the sample belongs to. As expected, L1 and L4 that have on average higher spacer number took higher computational time. Additional support to the impact of final CRISPR length onto computational time is the case of the L4 strain ERR751561: this strain harbors a rare large deletion letting it with only 25 DVR (**S7 Table**) and it showed a low computational time of 99s (L4 outlier highlighted by an arrow in **Fig 3**).

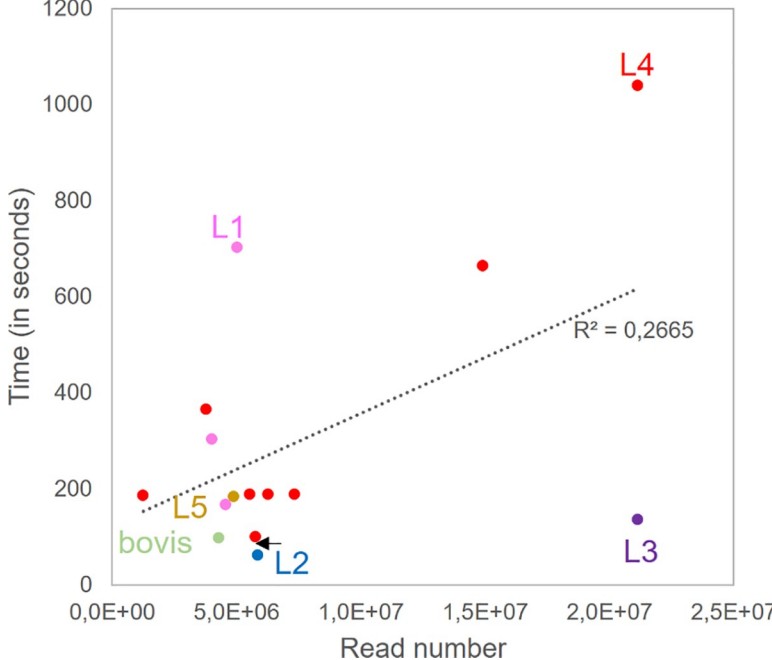

**Fig 3. CRISPRbuilder-TB swiftness on a representative sample set.** Pink = L1 samples; Blue = L2 sample; Purple = L3 sample; Red = L4 samples; Brown = L5 sample; Green = L6 sample.

## Discussion

We set up a fast semi-automatic pipeline called CRISPRbuilder-TB, available at https://github.com/cguyeux/CRISPRbuilder-TB to reconstruct CRISPR-Cas locus from MTC short reads sequencing runs. This tool is in the line of tools avoiding genome assembly to reduce computational time and potential biases such as PhyResSE, Snippy, etc. [27]. CRISPR loci are renown for the difficulty to reconstruct them. Existing tools have until now focused on their detection, an approximate reconstruction to enable comparisons in complex samples such as metagenomics data (S1 Table). Little attention has been brought to the exact reconstruction of CRISPRs. This exact reconstruction is necessary to infer evolutionary events affecting the CRISPR loci.

As CRISPR locus has been used for more than two decades in *Mycobacterium tuberculosis* complex as a first-line molecular epidemiology tool, we focused on this species. We reconstructed the CRISPR locus of more than 434 strains representative of all MTC diversity. With this first-hand sample, we showed here that MTC CRISPR underwent point mutations in both direct repeats and spacers, single DVR and multi-DVR duplications, IS insertions. We first discuss the robustness of CRISPRbuilder-TB, we then discuss its specificity in terms of adaptation to other species and adaptation to sequencing methods other than short-reads ones, and then comment on its advantage compared to other CRISPR identification tools.

### Robustness of CRISPRbuilder-TB in reconstructing CRISPR loci

A first evidence of the robustness of CRISPRbuilder-TB was its ability to reconstruct the 83 simulated MTC CRISPRs. These simulated CRISPRs included random evolutionary events among which point mutations in spacer or in direct repeats, DVR duplications, IS insertions. For all tested read length (75 to 300 bp) and all read depths (between x10 and x200), we obtained perfect loci reconstructions.

A second evidence of this robustness was CRISPRbuilder-TB ability to reconstruct the CRISPR loci of complete reference genomes. For these samples, we had both SRR and complete genome sequences deriving from Sanger sequencing. The reconstructed CRISPR loci proved 100% concordant with the complete genomes, except for H37Ra. For this sample, CRISPRbuilder-TB likely corrected an error in the available sequence: the complete genome exhibits unlikely dinucleotide repetitions, repetitions that do not exist in its closely relative H37Rv.

A third evidence of the robustness of our pipeline relies in the profiles derived from unknown SRR: for the 434 CRISPRs we reconstructed, we obtained CRISPR profiles in full agreement with the lineage/sublineage assignations and/or clustering derived from SNPs. These agreements concern standard DVR deletions, but also IS*6110* insertions, spacer or DR variations, or duplications. For instance, we observed the DVR43-50 deletions in all L4 samples, we observed an IS*6110* insertion downstream of spacer 41 in all 4.1.2.1 samples, a variant in sp4 in all L6 samples, a tandem duplication of DVR5 was observed in all L2 and L3 samples still harboring this region of the CRISPR (L2.1, most L3), etc. (see also S7 T). We could also confirm all specificities identified in the pioneer work of van Embden *et al.* using targeted Sanger sequencing such as DVR35 duplication for all samples outside L4 and most samples of L4, DR variants between sp30-31, sp 50–51, sp64-65, sp66-67 and 67–68 [14].

What may limit the robustness of CRISPRbuilder-TB? The first limitation is of course reads quality. We therefore recommend to check that average quality is high by running standard tools such as PhyResSE [27]. Another important limitation can be the presence of mixture: several genomes mixed in the same SRR. Our tool allows ignoring very low signals due to low level contaminations, but it is not adapted to handle mixed samples. To make sure that the

samples are not mixed it is recommended to run them with tools such as the one developed by Sobkowiak et al. [28].

A third limitation to robustness of CRISPRbuilder-TB is read length, but with a minimal length that is relatively low and largely achievable using widely used methods: reads need to include at least 15bp of consecutive spacers in addition to the DR. This makes a minimum read length of 66bp. We did however not confirm robustness below 75bp.

CRISPRbuilder-TB reconstructions result in a high level of additional information as compared to existing methods reconstruction MTC CRISPR diversity: both *in vitro* and *in silico* spoligotyping only deal with the presence or absence of specific spacers, with methods tolerating non-fully concordant sequences. As a result, all the information of spacer and DR variants, DVR duplications and IS*6110* insertions is lost when using the former methods, however not when using ours.

## On the specificity of the proposed tool

CRISPRbuilder-TB is, as its name makes it explicit, specific to the *Mycobacterium tuberculosis* complex. Could it be easily adapted to other species? Yes, CRISPRbuilder-TB could easily be adapted to other models and we indeed plan to propose versions adapted to other species. Independent developments using the current method are also encouraged. To apply CRISPR-builder-TB to other species, an important step is the set-up of a catalogue of remarkable sequences as depicted in **Fig 1** and described in detail in Section 4d. A possible procedure to do so is: (1) to retrieve the DRs and a first set of spacers of the species using CRISPRCasFinder [29], (2) to retrieve the patterns flanking the CRISPR on one genome, and (3) to enlarge the catalogue each time a new spacer or DR is found. If insertion sequences are likely to insert in the locus, they can be directly included in the catalogue. Another possibility is to let the pipeline identify them, as any sequences bordering the spacer contigs will be listed in the output.

What about the ability of CRISPRbuilder-TB to handle other raw data than short-reads sequences? If this method has been set up for current short read sequencing methods, it is fully compatible with longer reads deriving for instance from PacBio or MinION sequencing technologies. Of course, if the locus is smaller than the length of the reads, the algorithm will have little relevance: it will be reduced to the alignment of the reads matching the sequences of interest, the extraction of the consensus and the translation of the sequences into remarkable sequences tags. In that case, CRISPRCasFinder with its user-friendly interface may be more suitable [29]. However, it should be noted that in the MTC alone, we can have up to a hundred spacers, counting duplicates, and up to 3–4 IS in the CRISPR, thus up to 15000 nucleotides. Reads have to be really long to uncover complete loci.

We also want to highlight here that, even if other tools might prove more suited to deal with long-reads technologies and if these technologies are probably the future of genomics, it would be regrettable to ignore the past. For MTC alone, there are more than 70,000 SRR available, most acquired using short-reads sequencing methods. It is likely that, within the next 5 to 10 years, the richness of these databases will continue to be exploited. CRISPRbuilder-TB may all the same be applied on other species/samples for which short reads data continue to accumulate.

## Comparison with existing tools

Most of the available tools to explore CRISPR loci such as PILER-CR, CRT, or more exhaustive and user-friendly tools that use these methods such as CRISPRCasFinder, analyze only assembled genomes [16,20,29]. Assembly of CRISPR is however very tricky as discussed below, so that this first step may not provide the true sequence of the studied organisms. Few tools use

SRA data, and to the best of our knowledge, only Crass, CRTmeta, CrisprDetector and meta-CRISPR are compatible with such data [21–23,30].

Crass was the first tool that could reconstruct CRISPR loci from raw sequence data [21], was recently updated, and we easily implemented on several runs, at least when these runs had reads of sufficient lengths. However, when applied on simulated CRISPRs, we found that Crass could not detect DR and spacers in reads of 75bp. We also showed that when successfully applied, it was very sensitive to IS insertions, and to duplications, and never recovered the complete set of spacers. In contrast, CRISPR-builder TB could retrieve IS*6110* sequences insertion positions as shown on strains for which reliable complete genome sequences were available (for instance H37Ra, L4.9, **S3 Table, S2 Text**). On other SRR, CRISPRbuilder-TB retrieved between 0 and 2 copies of IS*6110* in the CRISPR locus, in either orientation (**S7 Table**, sheet "IS", and see also companion article Refrégier et al [1]). The tendency of Crass tool to miss duplications and/or insertions and the neighbouring spacers was further confirmed when applying it to MTC SRR (for instance, on ERR025424, lineage 4.1.2.1, coverage = 555.97, reads length = 116, 8/38 spacers were detected, moreover with errors in their sequences; ERR751335 from L5, **S2 Text**).

Other tested softwares included CRISPRdetector and metaCRISPR. CrisprDetector is a generalist tool: it does not imply any a priori knowledge of DRs or spacers, but that gives out only the DR sequence [22] as shown in **Table 1**. Other drawbacks include, it being based on an obsolete version of Python (version 2), its slowness and its high RAM demand (for instance, ERR025424, 21143954 reads, it took over 45 GB of RAM). Concerning metaCRISPR, we did not manage to make it run despite several trials.

## On the use of CRISPRs from complete genomes in public databases

The systematic study of the MTC CRISPR loci of the complete genomes highlights some errors in CRISPR public data that may be true for other species. We will now remind of the main pitfalls that we identified in this species and provide some advices on how to deal with other species.

By exploring MTC complete genomes deposited in public databases, we first showed that reference samples had high quality CRISPR reconstructions, likely due to partial or full Sanger sequencing. Conversely, another relatively large proportion of the ~200 complete genomes (more than 1/3) showed a clearly problematic locus, not trustworthy at all. This does not mean that there is no benefit in sharing such data, which can be informative for the rest of the genome. However, the problem is that it is difficult to know *a priori* whether, for a given genome, the CRISPR locus is, or is not, trustworthy. And this kind of problem may be true for species outside MTC.

The most likely reasons for this average low quality of CRISPR information is the difficulty to deal with this complexity when explored using short reads sequences. Obviously, a number of MTC studies have failed to do so. The difficulty of such a reconstruction, and the errors that result from it, have their source in several causes, some of which have already been introduced previously.

First, the CRISPR locus is by nature a very difficult area to assemble *de novo* based on short reads, at least with the same parameters as those that are efficient for most of the other sequences. The main pitfall is repeated sequences either due to the nature of the locus, insertions or to duplications during its evolution. Indeed, De Bruijn approaches look for an Eulerian path in the graph whose vertices are the k-mers, and for which there is an edge between two vertices if, and only if a suffix of one is a prefix of the other. In MTC, CRISPR locus contains a high number of DRs, IS*6110* insertions, spacers that sometimes share similarities (the

beginning of spacer 33 is the end of spacer 36, for example), multi-DVR duplications. All these events lead to possible bifurcations in the graph. In addition, the assembly is usually done by Velvet with its default maximum k-mer size of 49 [18]. In the best case scenario where this size has been set to its preconfigured maximum, knowing that a DR is size 36, this leaves only 13 bp of overlap to be shared between the two spacers, upstream and downstream, which multi- plies the incorrect bifurcations in the graph. Increasing this limit value requires recompiling Velvet from its sources, which obviously is not a common place, and only a few or no people who submitted their assembled *M. tuberculosis* genomes as complete genomes have done.

Second, if not relying solely on *de novo* assembly, genome assembly from SRR is often built by alignment to a reference genome. In our case, most assemblies used H37Rv, the first sequenced and well-studied L4.9 reference strain. In addition to having a CRISPR similar only with its closest relatives, this strain has specific features that increase its inability to help recon- structing other CRISPRs: it harbours several deletions, it has no duplication, and only one IS insertion, the ancestral IS*6110* copy upstream of spacer 35. When mapping reads to this refer- ence, samples containing spacers not present in H37Rv (such as sp30-31 or sp43-50) are likely to be discarded or misplaced. This is likely why the majority of the spoligotypes derived from the complete genomes available on the NCBI appeared to be L4-9 like, while at the SNP level, the lineages found were more diverse.

In conclusion, in this article, we have shown why CRISPR loci should not be assembled using standard tools and we have begun to reveal the unexpected diversity of MTC CRISPRs. This was made possible thanks to CRISPRbuilder-TB a new semi-automatic method that allows, for runs with a reasonable genome coverage and read size, to reconstruct CRISPR-Cas locus in a reliable, fast and robust way. It revealed duplications of various length, variants of spacers and DRs, and insertions of IS*6110* sequences, *i.e.* a full range of evolutionary events that may be found in other CRISPR loci, but no insertion of new spacers. In a companion arti- cle, we describe the high diversity of MTC CRISPR locus unveiled by our new method, we establish a list of notable elements by lineage, and infer MTC CRISPR various mechanisms of evolution, discussing for instance the likely non-functionality of spacer acquisition of this locus [1]. Among our objectives is the transformation of our tool into a professional quality software, so that the whole community can benefit from it. We also wish to study each lineage separately and in depth, on large sets of representative genomes, in order to reveal the fine evo- lutionary dynamics of the CRISPR-cas locus.

## Design and implementation

### Sequence data

A first set of data is constituted by seven reference clinical isolates, for which both complete genomes and short reads sequencing runs were available, downloaded from the NCBI website, and renown as reference strains (**S3 Table** and **Fig 1**). This selection was made with the a priori that these complete genomes would be highly reliable. This concerns the following strains: CDC1551, Erdman, F11, H37Ra, W-148, and the *M. bovis* BCG str. Pasteur 1173P2 and Tokyo 172.

A second set of data concerns all non-reference clinical isolates for which complete genomes were available at the time of the study (but for which no short reads sequences were available).

The third set of data comes from a collection of sequence reads archives from NCBI and from European Nucleotide Archive (ENA), that has been retrieved from some state-of-the-art articles to represent the diversity of MTC lineages [9,31–34]. This collection was completed by SRA queries on the NCBI search engine, with taxid values of 33894 and 78331, corresponding

respectively to *M. tuberculosis variant africanum* and *M. canettii* organisms. The names of SRA run accessions (SRR) were compiled, then the actual WGS sequencing data were automatically downloaded via the fastq-dump command of the sra-tools package. This led to a database of about 3,500 runs in the form of reads. This database is meant to be a good representative of MTC diversity, both at the lineage level and regarding geographical origins. A first selection on these runs was carried out, first of all concerning the sequencing technology, which should have been paired-end Illumina to avoid having to manage different formats in our scripts. We also recovered the size of the reads and the average coverage, and discarded all runs, either corresponding to weak covers (<50x) or with too short reads (minimum size of reads: 75 bp). This collection, once cleaned, was automatically annotated using the "Annotation" script described below, in order to attribute to each run its lineage, its spoligotypes "*old format*" (43 spacers) and "*new format*" (68 spacers for MTC, 98 spacers including *M. canettii*), as well as its Spoligo-International-Type (SIT) as described in [35]. A subcollection of 434 SRA samples representative of the global genome diversity was further analyzed using CRISPR-builder-TB (see "Runs' additional selection" below).

**Runs annotation.**    As a first annotation of the short sequencing runs (WGS data), we assigned the lineage/sublineage, for each single nucleotide polymorphism (SNP) referenced in [6] for all lineages, in [7] for L4 sublineages, in [9] for L2 sublineages, and in [8] for L1 sublineages. The annotation was made automatic by a script written in Python language that extracts, from its position in the reference genome H37Rv, a neighbourhood of 41 nucleotides centered around each SNP. For each run and each lineage-defining SNP, this 41 nt sequence was then blasted on the read sequences (blastn, maximum e-value 1e-5, from a local blast database calculated for each genome). At each blast output, we then counted the number of matches that contain the 41 nt version of H37Rv, and the number of matches that contain this pattern whose central nucleotide has been replaced by the SNP tabulated in **S2 Table** (SNP sheets). If the number of mutated units was significantly higher than that of reference H37Rv, the line associated with this SNP was then added to the genome considered.

As a second annotation, we provided the *in silico-derived* old and new formats of spoligo-types based on the presence/absence of known spacers. It follows the same principles as Spolpred [17]. We blasted each spacer on each of the read sequences (blastn, e-value < 1e-6), and we calculated the number of matches for each spacer (without looking at whether the sequences matched exactly, as spacers could have been mutated): if this number of matches exceeded 5% of the mean genome coverage, then we considered that the spacer could be added to the spoligotypes. At this level, the percentage has been preferred to a simple occurrence, because, for a certain number of runs, some spacers appeared in 2 or 3 reads when the number of occurrences of the other spacers exceeded, e.g., 70 –and this phenomenon tended to increase with coverage. These few spacers must obviously correspond either to a contamination, to a very minor strain in a likely double infection, or less likely to similarities that appeared by chance due to reading errors, the latter increasing with the number of reads. As for the threshold value for the percentage, it was set in this way after various tests, and by comparing the spoligotypes produced with those known for reference strains. The SIT could then be deduced from a correspondence table derived from SITVIT2 [35], however keeping in mind that experimentally- produced are not always 100% congruent to *in Silico*-produced spoligotypes.

**Complete genomes annotation and analysis.**    Scripts adapted from those set-up for SRR were written to extract information from complete genomes: 43-spacers spoligotype profiles, lineage/sublineage assignation according to SNPs. MIRU-Profiler was used to infer MIRU types from complete genomes [36]. Resulting patterns were entered in the on-line tool

TBminer (http://info-demo.lirmm.fr/tbminer/index.php) to identify most likely MTC lineage and sublineage assignation according to MIRU-VNTR, spoligo profile, or their combination [26].

**Listing of CRISPR-Cas remarkable sequences.   Direct repeats and spacers**

As each spacer lies between two DR sequences, and each DR lies between two spacers, we planned to detect new spacers using known DRs, and new DR variants using known spacers to build an exhaustive "catalogue of remarkable sequences". We started it with published sequences. DR0 is the name given for reference DR [14], reference spacers k are referred to as $esp_k$ (**S2 Table**).

We then looked for spacer variants, using regular expressions in randomly picked up runs from the sample #3 database. More specifically we searched in all the reads for patterns made up of: the last 12 nucleotides of the DR0, followed by a variable sequence with a size between 10 to 70 nucleotides, followed by the first 12 nucleotides of the DR0 (findall method of the python re module, patterns: DR0[-12:]([ATCG]{10,70})[DR0:12] and its reverse complement). The subsequences thus produced were then compared to the reference spacers as soon as they exceeded a number of occurrences fixed according to the coverage: if a given subpattern frequently appears between these two sections of DR0, and if it is not part of the known spacers, then it is determined whether it is a new spacer or a variant of a known spacer, in the following manner. The known spacer most similar to the detected subpattern is looked for, using a Needleman-Wunsch editing distance (compatible with substitution, indels, and gap insertion operations). If this similarity is greater than 95%, the subpattern is considered to be a variant of the most similar spacer and is integrated as such in the catalogue with a label of the following type $esp_k(i)$ where $i$ is the variant rank; otherwise, it is integrated in the catalogue as a new spacer as $esp_l$ where l = previous spacer number +1, see Algorithm #1 in **S3 Text**.

We then use this enhanced catalogue of spacer variants to find DR variants, in the same way as above. For each pair of spacers espk, espl, for k,l = 1. . .98, we look in the reads for subunits consisting of the last 12 nucleotides of espk, followed by 30 to 40 nucleotides, themselves followed by the first 12 nucleotides of espl. Again, reverse complement was considered, to double the number of matches, and the possibility of a "\n" for reads spread over more than one line was also included. The new DRs obtained were then used in a second phase of discovery of spacer variants, as before, taking into account that the sequences bordering spacers can be variants of the DR0.

**Other sequences of interest**

To this collection of subpatterns to be discovered in the CRISPR loci, we added:

1. the beginning and end sequences of IS*6110* and its reverse complement (40 bp each time).

2. CRISPR approximate borders: sequences corresponding to Rv2816c (*cas2* gene of the cas locus) and Rv2813c, reputed to border the CRISPR locus [10].

3. CRISPR exact-flanking sequences: the reads including the end of the *cas2* gene have been extracted from a small collection of genomes from the database presenting spacer 1 in its "new spoligotype" to retrieve likely ancestral closest border to CRISPR locus. The consensus sequence located downstream has been reconstructed, then the reads including the latter were recovered. These included a DR0 followed by the spacer "new" number 1. After verification (blast), this CRISPR-flanking pattern was indeed found in a large set of genomes in our collection, so it was added as such to the catalogue of patterns of interest. The same treatment was performed on genomes with spacer 68 to identify the end sequence between the latter spacer found in MTC stricto sensu (without *M. canettii*) and the Rv2813c gene. The corresponding pattern was also added to our catalogue (see **S2 Table**).

**Locus reconstruction.   Contiguage**

For each run, the sequences of interest mentioned above were first blasted on all the reads (blastn, evaluated 1e-7), in order to extract the small set of reads potentially covering the CRISPR locus. This small set of reads was then extended, where each read of size n was transformed into its n-k+1 k-mers, where k is equal to the integer part of 4n/5. The next step, is inspired by a classical contiguage by De Bruijn approach [37]. On the one hand, it was carried out to have a good coverage of CRISPR in terms of k-mers, and this even if the original coverage was not higher than 50x. On the other hand, in order not to definitively disqualify for the next steps a full read with a possible reading error, only its k-mers containing this error will be disqualified. Corresponding algorithm is available in **S3 Text** (algorithm #2).

A sequence is thus randomly extracted from this set of k-mers potentially covering the CRISPR, serving as a starting point for the first contig, to which an initial score of 1 is associated. The k-mers such that their first k-1 nucleotides correspond exactly to the last k-1 nucleotides of the current contig are then obtained from the set of k-mers. It is then regarded if the majority of the latter have the same last nucleotide (i.e., in position k). If this is the case, this nucleotide is added to the current contig, the k-mers that have matched are removed, their number is added to the score of the current contig, and the progress continues to be made in the reconstruction of the locus with the next nucleotide. If this is not the case, we start again with the other side of the current contig, looking for k-mers whose last k-1 nucleotides correspond exactly to the first k-1 nucleotides of the current contiguous. And the latter is no longer extended from his tail, but from his head.

At a time when no consensus seems to be emerging for the new nucleotide to be added to the current contig, this latter is stored separately with its score, and the whole process is repeated from a new randomly extracted k-mer. As, at each iteration, at least one k-mer is removed from the original set, this process has an end, leading to a more or less long list of potential contigs, themselves more or less long.

The contigs are then manually processed by decreasing score, in order to reconstitute the CRISPR structure. To this end, the catalogue of sequences of interest (variants of spacers and DRs, sequences bordering the IS*6110*, and the start and end patterns of the locus) is iterated, in order to replace each nucleotide sub-sequence by its name using the replace method of the str class (python). The result of this post-processing of the previously obtained contigs is a reasonably sized character string, including patterns of the form *spX(Y)* for the variant Y of the spacer X, *DRX* for variant number X of the DR, as well as the words *begin_IS*6110*, *end_IS*6110*, *begin_IS*6110c*, *end_IS*6110c*, *starting_pattern*,*ending_pattern*, *Rv2816c*, and *Rv2813c*. This translation makes it easier to understand the contigs obtained, and makes it easy to detect a break in the order in which the spacers appear. It also allows to detect new variants that had not been detected until now, and to add them after naming to the database of remarkable sequences. In the vast majority of the cases studied manually (but read exceptions in Duplication paragraph below), one to three contigs depending on the number of IS*6110* insertions in the locus (those with the highest scores) were sufficient to reconstruct the entire locus. The extreme elements of said contigs always were either the sequences bordering the locus or a beginning or end of IS*6110*(c).

**Duplications resolution**

If the reconstruction, mentioned above, of the CRISPR locus makes it possible to highlight the tandem duplications of spacers, in the case of read files of size >75 (leading to k-mers >56 bp, as in our selected WGS data), it nevertheless passes through possible duplications spread over several spacers. Let us suppose that we have a pattern of the form: spk*spk+1*...*spl*spk +1*...*spl*spm. Then, once the contig is rebuilt to the end of spacer number l (and its DR),

what comes next in the reads concerns both spk+1 and spm: when these two sequences diverge, there is no longer a nucleotide consensus in the considered reads, and the expansion of the contig stops. In addition, the k-mers of the second repeated pattern were used in the expansion of this contig when it was at the first pattern, to a number of k-mers used and removed twice as large as expected, and to the impossibility of reading the repetition of the pattern.

At this stage, we can conclude that if the expansion of a contig has not stopped on an IS or a sequence bordering a CRISPR locus, and if the score of said contig is higher than expected, then there is a suspicion of large-scale duplication. To resolve this situation, post-treatment was added to the locus reconstruction pipeline: for each pair of spacers (k,l), k,l = 1. . .98, we count the number of k-mers containing the last 12 spk nucleotides, followed by any of the DR variants, followed by the first 12 spl nucleotides. And couples whose number of matches is significant are displayed in lexicographic order. In this list, a pattern of the form spl*DRX*spm, l≥m, is proof of a duplication (in tandem when l = m): after l, we loop back to m<l. Of note, the successions of spacers involved in this duplication have a number of k-mers of the order of twice the successions of spacers located outside this duplication. And this doubling of the number of matches is a form of cross-validation of the duplication.

At this stage, we are therefore able to reconstruct the entire CRISPR locus from Illumina paired-end reads, provided that the coverage and size of the reads are reasonable, and this by being able to detect duplications, spacer and DR variants, and IS insertions. This process is 95% automated (https://github.com/cguyeux/CRISPRbuilder-TB), but it requires human intervention to finalize the assembly of the contigs (**S1 Fig**). Once this locus has been reconstructed, the resulting spoligotypes (old and new) can be compared to spoligotypes based on presence/absence of spacer sequences.

**Runs' additional selection**

A final point remains to be clarified at this stage, namely how the WGS runs used here for CRISPR reconstruction were selected from our database of ~3,500 items. Indeed, although much of the reconstruction has been automated, the remaining 5% takes a little time to be properly carried out. Not wanting to waste time rebuilding loci where nothing has happened, in terms of insertion and duplication, we have taken part of the pipeline detailed above to make a selection of the runs of interest. These correspond to samples carrying duplications as well as samples carrying IS*6110* insertions.

For a given run, we focus on reads returning matches during a blast on sequences of interest (DR and spacers). This again is performed using k-mers derived from the reads as described above. Then, patterns of the shape of an end of spacer l, followed by a variant of DR, itself followed by a beginning of spacer m, where l≥m, are looked for, as they are signs of duplication. Similarly, patterns of the form end of spacer k, followed by 0 to 36 nucleotides, themselves followed by the beginning of IS*6110*, are looked for insertions in DRs. Finally, ends of DR variant, followed by a certain number of nucleotides, and then the beginning of IS*6110*, are searched for insertions in spacers (with all possible variations in terms of layout and reverse complement). Only runs with either of these conditions were further considered, as basis of knowledge for the numerical study detailed below.

**MTC CRISPR simulation.**   In order to explore the ability of our method to reconstruct new CRISPR locus, we simulated the evolution of a locus subjected to genomic recombinations and point mutations, retrieved the composition of several evolved CRISPR loci, simulated artificial reads for each of them, applied our reconstruction technique, and compared the reconstruction to the actual composition of the evolved CRISPR.

The locus on which evolution was applied possessed all the spacers and the IS*6110* at his standard position (between spacers 34 and 35). Starting from this original locus, we

programmed an iterative evolution. At each step, we programmed the following evolutionary events according to a probability draw: 1) tandem duplications of spacers; 2) new insertion of the IS*6110*; 3) possible IS*6110* recombination leading to the deletion of embraced sequences, once at least two ISs were in the locus. The probability that such events occur were set respectively at 10%, 5%, 5%. For evolutionary events 1 and 2, uniform laws controlled the spacer subjected to duplication or next to which the IS*6110* inserts, if any. In addition, 5 punctual mutations were included for each evolutionary step. The evolution was performed using ten iterations.

Ten loci were simulated (each based on 10 independent iterations) and saved. Each evolved locus was then used to generate simulated sequencing reads, with a read size varying from 75 to 300 bp (75bp, 125bp, 200np, and 300 bp), and for an average coverage of 50x or 200x. No error linked to the simulated sequencing process was added.

A second set of 3 loci were simulated with adjusted frequencies of evolutionary events. Corresponding reads were generated using a homemade python program using random extraction of sequences of a definite length $l$ from a random position i until desired coverage is reached (items extraction from a list [i: i+$l$+1]). Targeted read sizes were of respectively 75, 125 and 300 bp, and the coverage was set at 10x.

The evolved loci were characterized in terms of total number of spacers, repeated DVR, number of IS insertions, deletions (**S1 Fig**).

From the reads, each simulated loci was reconstructed using CRISPRbuilder-TB method as well as other competing algorithms, and then compared to the corresponding evolved locus.

**Swiftness evaluation.**   To document the efficiency of CRISPRbuilder-TB, we made it run on a standard individual computer Intel Xeon E-2276M CPU @ 2.80GHz × 6, with 32 Go of RAM.

## Supporting information

**S1 Table. List of other tools to explore CRISPR presence.**
(XLSX)

**S2 Table. Table listing the DR, spacer variants and specifically searched patterns in our pipeline (page 1) where SNPs are highlighted in red, and Tables listing updated SITVIT listing and SNPs used to infer classification (page 2–6).**
(XLSX)

**S3 Table. CRISPR-Cas features of reference strains according to complete genome sequences and according to CRISPR-builder run on WGS-data.** CRISPR features derived from complete genome sequences using Spolpred-like tool.
(DOCX)

**S4 Table. Spoligotype and MIRU-VNTR patterns of non-reference complete genomes, derived classification and comparison with classification derived from their SNPs.**
(XLSX)

**S5 Table. Spoligotype and MIRU-VNTR profiles of Public complete genomes with discrepancies between SNP-based and spoligo-derived classification according to TBminer webtool [26].**
(XLSX)

**S6 Table. CRISPR-Cas locus profile reconstructed from public WGS runs and representative of MTC diversity.**
(DOCX)

**S7 Table. Extensive annotation of the selection of reconstructed CRISPRs using our pipeline (SRA, dataset #3, see Fig 2).** Legend IS6110 sheet. Column A: lineage assignation according to Coll et al., column B: Lineage and Sublineage according to Palittapongarnpim et al. for L1, Shitikov et al. for L2, Stücki et al. for L4; column C: spoligotyping-based alias name, column D: SIT nomenclature, column E: selection set, column F: Accession number, column G: common Strain name, column H: location, column I: spoligotype (old), column J to end: the first line designates the name of the gene or the spacer ID; the second title line designates the spacer ID according to the classical spoligotyping nomenclature (also visible by yellow shade). Vertical bars stand for IS*6110* copies within DR, in green for copies in orientation 1 (antisense) and red for orientation 2 (sense). Color boxes are for insertions within a gene or a spacer, with the same color for the orientation than above. The number in the box indicates the position (nucleotide) where the insertion occurred in the coding sequence. Finally, a large colored tube with white squares depicts a very likely recombination event between two insertion sequences that led to the deletion of all sequences between them (thus, only one IS6110 remains). "esp" sheet: column E-AD: spacers with at least one variant in the sample set. The SRA are shown in lines in the same orders as others sheets. Crosses show samples with a variant in the corresponding spacers. When several variants were found, they are numbered. "DR" sheet: columns F to AE list the variant DRs and their bordering spacers. A black square indicates a variant, a white square corresponds to the wt version, a dot corresponds to a deletion. "Dupl" sheet: DVRs duplicating in some samples are shown in the order in which they occurred, black squares indicating actual presence of the DVR at this position, white square indicate the absence of the DVR. Red blocks identify the duplicated DVR blocks.
(XLSX)

**S8 Table. DR and spacer variants for the representative set of MTC diversity.**
(DOCX)

**S1 Text. Simulated CRISPR-4,5,6 reconstructions by Crass, CRISPRdetector and CRISPRbuilder after reads simulation of 75bp in length, 125bp in length, 300bp in length.**
(PDF)

**S2 Text. Comparative Crass and CRISPR-TB results obtained on two genomes ERR751335 (L5) and SRR6407486 (L4.9).**
(PDF)

**S3 Text. Algorithms used to reconstruct CRISPR locus (spacer discovery, and contiguage).**
(PDF)

**S1 Fig. Visual Display of Simulated CRISPR-4,5,6 (CRISPR before evolution, simulated CRISPR, simulated SRAs, CRASS output, CRISPR detector output, CRISPRbuilder-TB automated output, CRISPRbuilder-TB automated output after visual inspection).**
(PPTX)

## Acknowledgments

CG, CS and GR are grateful to Gilles Vergnaud and Christine Pourcel (Institut for Integrative Cell Biology, I2BC, UMR9198, CEA-CNRS-Université Paris-Saclay, Gif-sur-Yvette, France) for helpful discussions at various steps of this project.

## Future directions

Future developments include the confrontation between CRISPR structure and SNPs from M. tuberculosis complex. Other potential developments could concern CRISPR building of other pathogens such as Clostridium difficile.

## Author Contributions

**Conceptualization:** Christophe Sola, Camille Noûs, Guislaine Refrégier.

**Data curation:** Christophe Guyeux.

**Formal analysis:** Christophe Guyeux.

**Investigation:** Christophe Guyeux, Camille Noûs, Guislaine Refrégier.

**Methodology:** Christophe Guyeux, Camille Noûs, Guislaine Refrégier.

**Project administration:** Christophe Guyeux, Christophe Sola, Guislaine Refrégier.

**Software:** Christophe Guyeux.

**Supervision:** Christophe Sola, Guislaine Refrégier.

**Validation:** Christophe Sola, Guislaine Refrégier.

**Visualization:** Christophe Guyeux, Guislaine Refrégier.

**Writing – original draft:** Christophe Guyeux, Guislaine Refrégier.

**Writing – review & editing:** Christophe Guyeux, Christophe Sola, Camille Noûs, Guislaine Refrégier.

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
