## [Decision Letter · Decision Letter 0]

26 Jul 2020

Dear Dr. Guyeux,

Thank you very much for submitting your manuscript "CRISPRbuilder-TB: “CRISPR-Builder for tuberculosis”. Exhaustive reconstruction of the CRISPR locus in Mycobacterium tuberculosis complex using SRA" for consideration at PLOS Computational Biology.

As with all papers reviewed by the journal, your manuscript was reviewed by members of the editorial board and by several independent reviewers. In light of the reviews (below this email), we would like to invite the resubmission of a significantly-revised version that takes into account the reviewers' comments.

We cannot make any decision about publication until we have seen the revised manuscript and your response to the reviewers' comments. Your revised manuscript is also likely to be sent to reviewers for further evaluation.

Sincerely,

Mihaela Pertea

Software Editor

PLOS Computational Biology

Mihaela Pertea

Software Editor

PLOS Computational Biology

Reviewer's Responses to Questions

**Comments to the Authors:**

Reviewer #1: The work presented by Guyeux and co-authors in this manuscript focuses on reconstructing the CRISPR locus of Mycobacterium tuberculosis complex strains from short reads sequences directly. The authors developed a bioinformatics pipeline to achieve this goal and presented their results compared to those obtained from other similar tools. A major strength of their work is that the performance of their pipeline (named CRISPRbuilder-TB) is evaluated by reconstructing CRISPR loci of simulated and complete MTBC genomes, where the structure of CRISPR loci are known. They show that 1/3 of assembled genomes contain mis-assemblies and erroneous CRISPR loci, and thus advocate for the use of their pipeline to reconstruct CRISPR loci from short reads directly. They also show how existing tools do not capture the full diversity at the CRISPR locus, including duplications, spacer and direct repeat variants, and locations of IS6110 insertions.

Major comments:

- The authors should include documentation on their project GitHub page (https://github.com/cguyeux/pyMTC/) explaining how the pipeline can be downloaded and run locally by providing examples, in addition to any software requirements.

- The authors provide a comprehensive list of available CRISPR identification tools. However, they only compared their tool (CRISPRbuilder-TB) to two other independent tools (Table 1). It is not cleat to this reviewer why only Crass and CRISPRdetector were used as comparison tools, whereas other cited CRISPR predictor tools (CRT, CrisprFinder, PILER-CR, CRISPRCasFinder, CRTmeta or metaCRISPR) were not. As some of these tools only accept assembled genomes as input, simulated reads could be assembled first and then assemblies used to run the remaining CRISPR prediction tools.

- Both in Table 1 and Table 2 is not obvious from the text provided that reconstructed CRISPR loci by CRISPRbuilder-TB are identical to simulated loci (Table 1) or reference genome loci (Table 2). A simple visual representation of simulated and reference vs. reconstructed CRISPR loci is needed to easily identify and compare regions of homology, order of CRISPR features (DR, spacers) and variants (SNPs, duplications, IS6110) between simulated and reference compared to reconstructed loci.

- Although, the authors have published a separate study on the evolution of CRISPR locus on MTBC, they should discuss whether there is evolutionary evidence that the CRISPR locus is functional. According to https://www.biorxiv.org/content/10.1101/2019.12.13.875765v1 , the CRISPR locus “of M. tuberculosis did not evolve by classical CRISPR adaptation (incorporation of new spacers) since the last most recent common ancestor of virulent lineages.” Would the authors interpret these results as a clear indication of a non-functional CRISPR locus?

Minor comments:

Abstract:

In ‘compared to more generalist tools’, please indicate here what these tools are here in the abstract.

Authors summary:

Line 55: ‘using short read sequences’ not ‘Short Reads Archives’

Line 56: ‘current assembled genomes’

Line 60: list here what existing methods CRISPRbuilder-TB was compared to

Introduction:

Line 67: indicate in brackets and include citations of the first two Mtb sequenced isolates

Line 73: given that spoligotyping is not taxonomically precise or fully ‘correct’, please replace ‘allows a correct taxonomical assignment’ with ‘allows an approximate taxonomical assignment’

Line 89: in more than a hundred thousand samples. Also include citation at the end of this statement.

Line 97: they have the same limitations as in vitro spoligotyping techniques.

Line 99. Use WGS abbreviation instead of ‘whole genome sequencing’, here and elsewhere in the text.

Line 99. By traditional de novo assembly tools

Results:

a. Description of CRISPRbuilder-TB

Line 129. A known limitation

Figure 2. In Dataset #1 Validation sample, do the authors used 7 complete (sequenced to completion) reference genomes? If so, they may want to change ‘Assembled genomes

collection from NCBI’ to ‘Complete genomes collection from NCBI’

In lines 130 to 132. A more explicit description on how CRISPR locus associated reads are extracted is needed here. From Figure 1 step 1, DR sequences are used to fish out CRISPR locus associated sequences from short sequence reads.

Figure 1 illustrates very well CRISPRbuilder-TB pipeline but lacks a legend in the manuscript. Figure 1 would also benefit from including what bioinformatic tools were used at each step.

Line 138. It would be helpful to report how many SNP differences (mismatches) the authors find among variants of the same spacer. This would inform the maximum number of mismatches in silico spoligotyping tools use to search for spacer occurrences in SRA data.

Lines 148 – 149. What do the authors mean by ‘translate these contigs in strings of words’? Do they refer to annotating contigs with CRISPR locus features such as DR, spaces, etc. If so the authors may want to use the term ‘annotation’ and rename step4 in Figure 1 ‘Translation into explicit contigs’ to ‘CRISPR annotation of contigs’.

Line 151. Related to the point above, the authors may want to use the term ‘Unannotated regions’ of contigs instead of ‘untranslated parts’.

b. Reconstruction of simulated CRISPR

Table 1. It is not cleat to this reviewer why only Crass and CRISPRdetector were used as comparison tools, whereas other cited CRISPR predictor tools (CRT, CrisprFinder, PILER-CR, CRISPRCasFinder, CRTmeta or metaCRISPR) were not. As some of these tools only accept assembled genomes, simulated reads could be assembled first and then assemblies used to run the remaining CRISPR prediction tools.

Table 1. As pointed before, it is preferable to use the term ‘non-annotated contig sequences’ than ‘untranslated sequences’

Line 163. Indicate how short (in bp) the shortest reads used by CRISPRbuilder were.

Line 163. As pointed before, use ‘annotation’ instead of ‘translation’.

Evaluation of CRISPR locus reconstruction based on WGS data of MTC reference strains

Line 187. The authors should stress here that the seven reference strains had been sequenced to completion, that is, they had an ungapped and complete chromosome sequence resolved. This is somehow implied with the term ‘reference’ genome.

Line 203. Do the authors mean CRISPRbuilder-TB reconstructed sequences were identical to those of the complete reference genome? Or those assembled de novo from short reads? Same point in Line 212 ‘assembled genome’.

Line 207. Citation needed right after statement ‘as described previously’.

219. Do the authors mean ‘public reference genomes’ by ‘public assemblies’ here? As pointed before, it is important to differentiate between complete reference genomes (ungapped, resolved and annotated) vs. de novo assemblies (made up of contigs).

d. CRISPR region in other MTC assembled genomes.

Line 230. In the statement ‘Most sequences seemed of high quality.’, what quality control metrics did the authors used to assess the quality of these 187 MTC public genomes? What sampling strategy was used to select these 187 MTC genomes? Are these complete genomes or assembled?

e. CRISPR evolutionary events in MTC

As pointed before, for line 138. It would be helpful to report how many SNP differences (mismatches) the authors find among variants of the same spacer. This would inform the maximum number of mismatches in silico spoligotyping tools use to search for spacer occurrences in SRA data.

f. Computational speed

Discussion.

Line 323. Use ‘complete reference genomes’ instead of ‘reference samples’.

Line 325. Use ‘concordant with complete genome’ instead of ‘concordant with assembled ones’

4. Material and methods

a. Sequence data

Line 480 to 481, replace ‘both assembled genomes and short reads sequencing’ with ‘both complete genomes and short reads sequences”.

Lines 486 and 487. What selection criteria did the authors followed to select these assembled genomes from public repositories?

Line 489. Do the authors mean European Nucleotide Archive (ENA) by EMBL-SRA?

e. Locus reconstruction

Contiguage

It is not clear to this reviewer why the authors did not make use of an established de novo assembler (such as Velvet or Spades) to assembly selected reads and, instead, came up with their own algorithm.

f. MTC CRISPR simulation

Lines 708 – 712. Can the authors specify what bioinformatics tool they used to simulate short reads?

Reviewer #2: This is a well written paper that provides important novel insight in the CRISPR genetic diversity and organisation of the locus and will help refine the MTC global population structure. It presents a detailed and comprehensive rational, approach and data for the problem. In my opinion, this is an important and worth while tool that should complement the current apps used for refining MTC phylogeny.

I have a few minor points for the authors to address.

Line 39: put comma after pipeline

Line 43: replace generalist with generalised

Line 89: replace hundred with one hundred

Line 90: replace to explore with exploration of

Line 124-126: this does not make sense. Please re-write

Line 159: replace Tool's with Tool

Line 180: Why is the data not shown? Please explain and provide data in suppl figs.

Line 286: replace Go with Gb

Line 362: Could it be easily adapted to other species? Re-write...not appropriate

Figures:

Largely fine, but would recommend modifying/edit text and font styles to look clearer and optimum.

Typos:

The manuscript is generally well written, but a thorough read-through is needed as there are a number of typos, formatting and grammatical errors. Some of which I have highlighted above.

Discussion:

I would like to see a little more about how this software has resolved the frequency and nature of IS6110 insertions in the CRISPR locus. Also, how does the software cope with large scale deletions of the locus due to IS6110?

**Have all data underlying the figures and results presented in the manuscript been provided?**

Reviewer #1: Yes

Reviewer #2: Yes

PLOS authors have the option to publish the peer review history of their article (what does this mean?). If published, this will include your full peer review and any attached files.

Reviewer #1: **Yes: **Francesc Coll

Reviewer #2: No
---

## [Decision Letter · Decision Letter 1]

24 Oct 2020

Dear Dr. Guyeux,

Thank you very much for submitting your manuscript "CRISPRbuilder-TB: “CRISPR-Builder for tuberculosis”. Exhaustive reconstruction of the CRISPR locus in Mycobacterium tuberculosis complex using SRA" for consideration at PLOS Computational Biology. As with all papers reviewed by the journal, your manuscript was reviewed by members of the editorial board and by several independent reviewers. The reviewers appreciated the attention to an important topic. Based on the reviews, we are likely to accept this manuscript for publication, providing that you modify the manuscript according to the first reviewer's recommendations.

Sincerely,

Mihaela Pertea

Software Editor

PLOS Computational Biology

Mihaela Pertea

Software Editor

PLOS Computational Biology

[LINK]

Reviewer's Responses to Questions

**Comments to the Authors:**

Reviewer #1: The authors have successfully addressed the following major comments:

- Detailed documentation on how to download, install and run their tool on GitHub has been now provided.

- They argued well why the used short reads as the tool’s input, and not assemblies, and thus the choice of CRISPR predictor tools they used to compare their tool’s predictions against.

- Added some discussion on the functionality of CRISPR locus in MTC.

However, the authors haven’t addressed the following point:

The newly added Supplementary_File3.odt, Supplementary_File4.odt and Supplementary_File5.odt do not provided “A simple visual representation of simulated and reference vs. reconstructed CRISPR loci is needed to easily identify and compare regions of homology, order of CRISPR features (DR, spacers) and variants (SNPs, duplications, IS6110) between simulated and reference compared to reconstructed loci.”

All minor comments were successfully addressed.

Reviewer #2: None

**Have all data underlying the figures and results presented in the manuscript been provided?**

Reviewer #1: Yes

Reviewer #2: Yes

PLOS authors have the option to publish the peer review history of their article (what does this mean?). If published, this will include your full peer review and any attached files.

Reviewer #1: **Yes: **Francesc Coll

Reviewer #2: No
---

## [Editor Report · Decision Letter 2]

8 Nov 2020

Dear Dr. Guyeux,

We are pleased to inform you that your manuscript 'CRISPRbuilder-TB: “CRISPR-Builder for tuberculosis”. Exhaustive reconstruction of the CRISPR locus in Mycobacterium tuberculosis complex using SRA' has been provisionally accepted for publication in PLOS Computational Biology.

Best regards,

Mihaela Pertea

Software Editor

PLOS Computational Biology

Mihaela Pertea

Software Editor

PLOS Computational Biology

---

## [Editor Report · Acceptance letter]

18 Dec 2020

PCOMPBIOL-D-20-00626R2 

CRISPRbuilder-TB: “CRISPR-Builder for tuberculosis”. Exhaustive reconstruction of the CRISPR locus in Mycobacterium tuberculosis complex using SRA

Dear Dr Guyeux,

I am pleased to inform you that your manuscript has been formally accepted for publication in PLOS Computational Biology. Your manuscript is now with our production department and you will be notified of the publication date in due course.

With kind regards,

Livia Horvath
